# A Methodology for the Assessment of Climate Change Adaptation Options for Cultural Heritage Sites

**Bethune Carmichael** [1,*]**, Greg Wilson** [2]**, Ivan Namarnyilk** [2]**, Sean Nadji** [3]**, Jacqueline Cahill** [3]**, Sally Brockwell** [1]**, Bob Webb** [4]**, Deanne Bird** [5] **and Cathy Daly** [6] 

1   Department of Archaeology and Natural History, Australian National University, Canberra 2601, Australia; sally.brockwell@anu.edu.au
2   Djelk Rangers, Maningrida 0822, Australia; gregwilsonmaningrida@gmail.com (G.W.); ivannamarnyilkmaningrida@gmail.com (I.N.)
3   Kakadu National Park Rangers, Jabiru 0886, Australia; sean.nadji@awe.com.au (S.N.); Jacqueline.cahill@awe.com.au (J.C.)
4   Fenner School of Environment & Society, Australian National University, Canberra 2601, Australia; Bob.Webb@anu.edu.au
5   Faculty of Life and Environmental Sciences, School of Engineering and Natural Sciences, University of Iceland, 101 Reykjavík, Iceland; Deanne.Bird@gmail.com
6   School of History and Heritage, University of Lincoln, Lincoln LN6 7WA, UK; CDaly@lincoln.ac.uk
*   Correspondence: bethune.carmichael@anu.edu.au

**Abstract:** Cultural sites are particularly important to Indigenous peoples, their identity, cosmology and sociopolitical traditions. The benefits of local control, and a lack of professional resources, necessitate the development of planning tools that support independent Indigenous cultural site adaptation. We devised and tested a methodology for non-heritage professionals to analyse options that address site loss, build site resilience and build local adaptive capacity. Indigenous rangers from Kakadu National Park and the Djelk Indigenous Protected Area, Arnhem Land, Australia, were engaged as fellow researchers via a participatory action research methodology. Rangers rejected coastal defences and relocating sites, instead prioritising routine use of a risk field survey, documentation of vulnerable sites using new digital technologies and widely communicating the climate change vulnerability of sites via a video documentary. Results support the view that rigorous approaches to cultural site adaptation can be employed independently by local Indigenous stakeholders.

**Keywords:** climate change adaptation; archaeology; cultural heritage; Indigenous; options analysis

## 1. Introduction

Climate change threatens to destroy hundreds of thousands of the world's cultural heritage sites [1]. Cultural heritage sites (hereafter 'cultural sites'), including historic monuments, archaeological sites and cultural landscapes, play a significant role in community identity, cohesion and sense of place, and this is particularly the case for Indigenous peoples [2], for whom cultural sites often provide a foundation for the maintenance of traditional cultural practices and behaviours. Australian Indigenous perceptions of climate change impacts on cultural sites are growing [3,4]. In northern Australia, Indigenous rangers from Kakadu National Park (KNP rangers) and the Djelk Indigenous Protected Area (Djelk rangers) have perceptions of sea level rise and more extreme storm surges increasing erosion of coastal middens and floodplain-fringing middens and rock art; more intense cyclones are perceived to be impacting coastal middens; and more extreme and frequent precipitation events to be eroding inland riverine rock art and contributing to the erosion of floodplain-fringing middens and rock art [3]. In response, KNP and Djelk rangers sought to adapt sites to climate change and build site

resilience [5]. To this end, they conducted a risk analysis at 126 rock art and midden sites (Figure 1) and prioritised sites at 'high' and 'very high' risk for adaptive actions [6]. Next, rangers sought to identify and appraise a range of adaptive actions for deployment at the prioritised sites. However, no methodology existed to aid them in this process: while some archaeological studies responding to climate change risks envisage a role for local stakeholders in site adaptation [7], none have considered a scenario in which they are central.

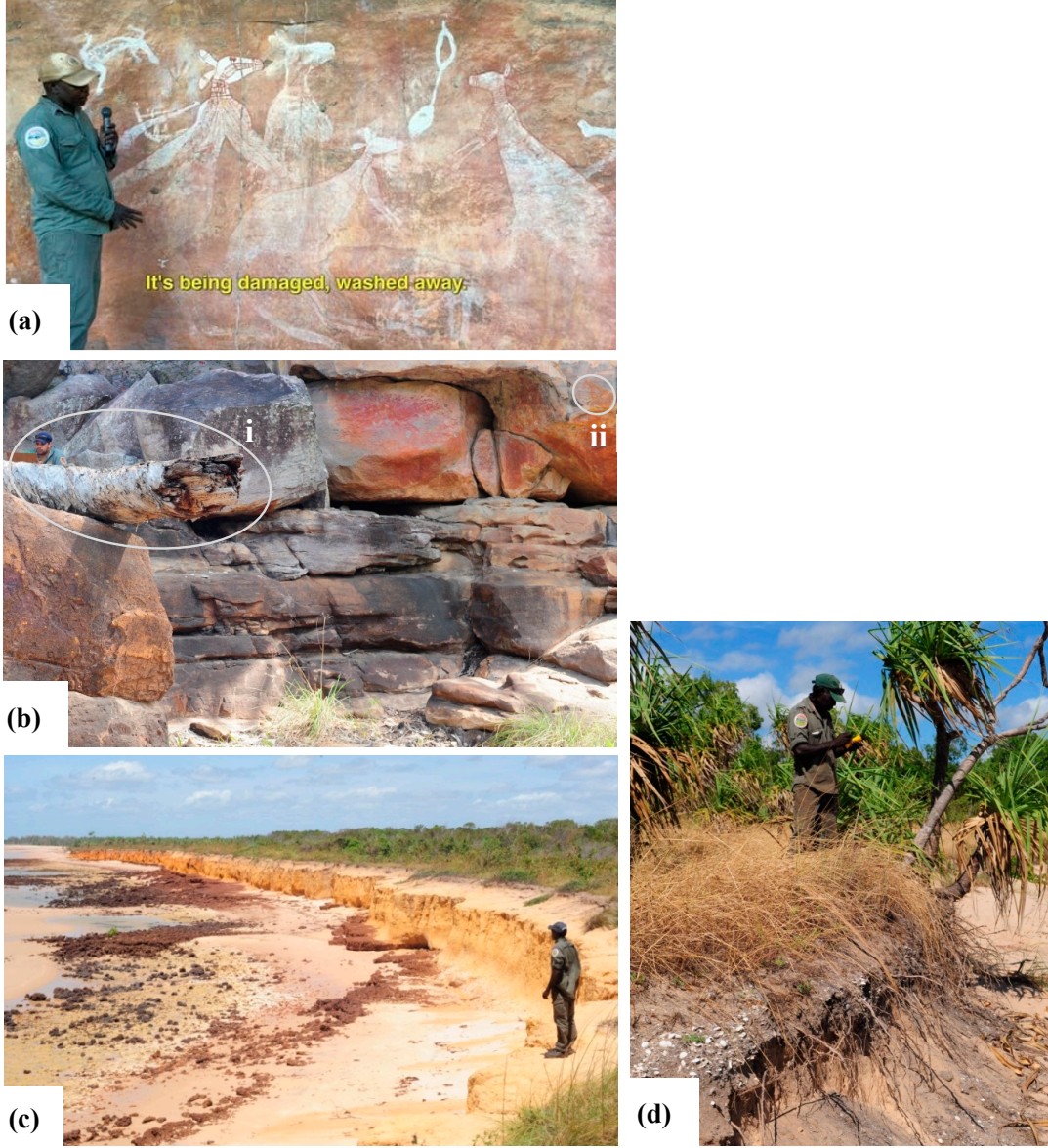

**Figure 1.** Assessing climate change threats to rock art and middens. (**a**) Djelk ranger Ivan Namarnyilk inspects flood damage to a rock art site close to the Cadell River, south of Kolorbidahdah. Screenshot from the documentary *Places in Peril: Archaeology in the Anthropocene* [8], a ranger-initiated communication tool aimed at conveying the gravity of the threat of climate change to cultural sites to a global audience. Viewable at link: https://vimeo.com/203773921. (**b**) KNP ranger Sean Nadji uses the risk field survey at a rock art site on the fringes of the Majela Plain. Flood debris (**i**) has been deposited on top of an outcrop adjacent to rock art (**ii**). (**c**) Djelk ranger Greg Wilson surveys coastal erosion. (**d**) Djelk ranger Greg Wilson records the location of an eroded coastal shell midden.

This study responded directly to this methodological deficit: it aimed to propose an options-analysis method for Indigenous ranger cultural-site adaptation and then determine whether it can elicit meaningful and comprehensive responses from rangers, and whether rangers have the organisational capacity to fulfil its requirements. When local stakeholders play a central role in cultural site climate change adaptation, adaptation plans gain political legitimacy and become responsive to local vulnerabilities and values [9]. All too often, local communities are the only stakeholders responding to the needs of cultural sites [10]. Tools designed to assist them adapt sites to climate change have the potential to build local adaptive capacity and cultural site resilience [11].

## 2. Materials and Methods

### 2.1. Case Studies

Two ranger groups participated in this study: the Kakadu National Park Rangers (KNP Rangers) are located in a national park administered jointly by the Australian government and Indigenous land owners, and the Djelk Rangers are based in an Indigenous protected area (IPA), the Djelk IPA, administered by an Aboriginal corporation (Figure 2).

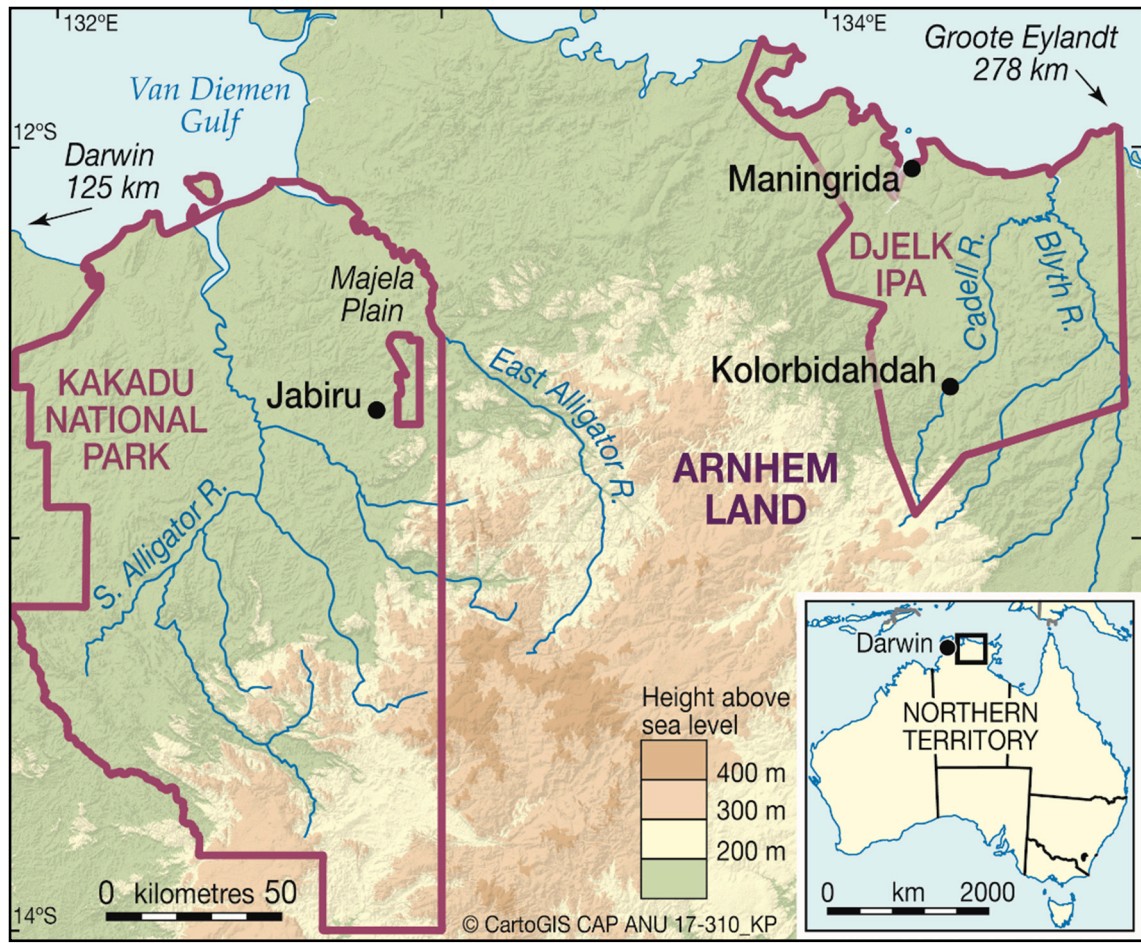

**Figure 2.** Location and setting of case study sites: Kakadu National Park and Djelk Indigenous Protected Area.

### 2.1.1. Kakadu National Park

Indigenous rangers are employed alongside non-Indigenous rangers across Australia's conservation estate, which includes national and state-administered parks. Indigenous rangers

undertake natural and cultural resource management, including fire, feral animal and weed management, cultural site maintenance and tourist liaison. Around a third of KNP rangers are Indigenous men and women. Their management practice draws extensively on traditional environmental knowledge, with a large proportion of Indigenous rangers being traditional owners of domains within the park.

Kakadu National Park covers 19,804 square km (approximately half the size of Switzerland) within the Alligator Rivers Region in the Northern Territory. Stage one of the park was inscribed on the UNESCO World Heritage List in 1981 for its natural and cultural values. While around 5000 rock art sites have been recorded, it is likely that 10,000 to 20,000 remain unrecorded [12]. Occupation has been dated from up to 65,000 years [13] and rock art reveals insights into Indigenous hunting, gathering, society and rituals from 28,000 years ago through to the present [14]. The park is administered by a Board of Management, comprising traditional owners but also Australian Government and non-Indigenous tourism sector representatives [15].

### 2.1.2. Djelk Indigenous Protected Area

In 2015, 70 IPAs across Australia covering some 63 million hectares of land [16] were managed by 108 Australian Indigenous ranger groups. IPAs require Indigenous landowners to nominate their estates for inclusion. These are subsequently recognised as part of the National Reserve System and attract government resourcing. Indigenous rangers undertake natural and cultural resource management, including fire, feral animal and weed management.

The Djelk IPA covers over 14,000 square km of land and sea country. During the wet season, Maningrida, the main township in the Djelk IPA, is inaccessible by land transport. It too represents exceptional natural and cultural value, including 12 separate language groups. Apart from archaeological studies on the Blyth and Cadell rivers several decades ago [17–20], cultural site documentation has been limited. The Djelk Rangers began operating in 1991, with the IPA being declared in 2009. Djelk employs over 30 men and women full-time, almost all of whom are traditional owners. As a subsidiary of Bawinanga Aboriginal Corporation, Djelk is directed by an Indigenous executive committee, which ensures that all decisions respect Indigenous cultural protocols. Djelk receives funding under the Australian Government's 'Caring for our Country' initiative [21].

### 2.1.3. Case Study Selection

In initially selecting the two groups, five ranger groups were approached: two from central Australia, the Warlpiri Rangers and the Tjuwampa Rangers; and three from northern Australia, Djelk Rangers, Gundjeihmi Rangers, and a cohort of Indigenous rangers from Kakadu National Park, some of whom were also Njanjma Rangers.

Rangers were approached either by non-Indigenous support staff or via a paid Indigenous consultant. Informal, face-to-face unstructured interviews centred on perceptions of climate change, impacts on cultural sites and whether planning for these impacts was a priority need. In central Australia, the Warlpiri Rangers were a relatively new and inexperienced group with limited perceptions of climate change and no perceptions of impacts on cultural sites. Tjuwampa Rangers had perceptions of increased temperatures resulting from climate change, but the perceptions of impacts were limited to increased feral animal damage to sacred trees as a result of an increased need for shade. Neither group rated climate change adaptation planning as a priority need. Central Australia's semi-arid deserts experience very extreme natural temperature and rainfall variation. Differentiating anthropogenic climate change from natural variation is possible by way of long-term climate trend data [22], but is difficult to perceive by those managing natural resources over the short to medium term. Of the three northern Australian groups, the Gundjeihmi Rangers were only newly established one; all members were under the age of 25 and unfamiliar with the concept of climate change.

In contrast, the KNP and Djelk Rangers were familiar with the concept of climate change, had strong perceptions of climate change impacts on cultural sites and considered managing these

impacts a priority [4]. They are long-established and highly experienced groups, responsible for maintaining rock art, midden and burial sites close to tidal and riverine flood zones.

An options-analysis methodology was derived from a review and synthesis of options-analysis methods outlined within five generic climate change adaptation planning guides [23–27]. These five were selected on the basis of their positive ratings against a set of assessment criteria by Webb et al. [11].

### 2.2. Option Identification

In line with a method commended by three sources [23,26,27], a preliminary options list was derived from the results of the preceding scoping and risk analysis (see Table 1, Item 1.1). During both steps, rangers regularly proposed adaptation options. An analysis was therefore then conducted of transcripts from workshop discussions, semistructured individual and small group (two to three participants) interviews, audio recordings of participant observations, and field notes, all accumulated during these steps. Data were analysed and grouped into themes with the aid of NVivo qualitative data analysis software.

In order to generate a comprehensive list of possible climate change adaptation options as a point of comparison for those identified by rangers, a literature review of archaeological studies addressing climate change adaptation was conducted. Past studies addressing the issue of climate change impacts on cultural sites have been largely concerned with risk assessment, while only a limited number considered adaptation options. We identified six studies with substantive consideration of adaptation options for cultural sites facing climate extremes, variability or change [7,28–32] (see Table 2 in the Results section). A number of other studies considered options for cultural site adaptation but only in passing, and so were not included [33–35].

**Table 1.** Comparison and selection of options-analysis methods. A plus (**+**) indicates that the method features in a given generic adaptation planning guide. A tick (**√**) indicates that the lead author adopted the method for the Djelk and Kakadu Ranger workshops, during which rangers endorsed its use. A cross (**✗**) indicates that a given method was not adopted by the lead author, and comments indicate why. Method 2.5.1 sets out the assessment criteria used, their source in climate change adaptation literature and the resulting questions posed to rangers. In line with the method commended by three sources [24–26], workshops began with a brainstorming exercise in which participants were invited to add additional options to the preliminary list.

| | | Generic Adaptation Planning Guides | | | | | Methods Pre-Selected for KNP/Djelk Workshops | |
|---|---|---|---|---|---|---|---|---|
| | | Burton et al. (2005) | Dazé et al. (2009) | Hinkel et al. (2013) | UKCIP (2017) | Willows et al. (2003) | | |
| **1.** | **Methods for *identifying* options** | | | | | | | **Comments on rejected steps** |
| 1.1 | Use options suggested during scoping/risk analysis. | + | | + | + | | √ | |
| 1.2 | Use a generic list of options. | | | | + | | ✗ | No generic list exists for cultural sites. |
| 1.3 | Use free brainstorming. | | + | + | + | | √ | |
| 1.4 | Use prompts to elicit options, i.e., | | | | | | | |
| | (a) Option qualities: i.e., 'low regrets', 'flexibility', etc.; | | | | + | + | ✗ | Better addressed during 2. Methods for appraising options (below). |
| | (b) Options addressing limits of existing program/strategy; | | + | | | | ✗ | Existing programs assessed as inadequate during project scoping: option responses captured at 1.1 (above). |
| | (c) Options to build adaptive capacity of stakeholders. | | | + | | | ✗ | Not comprehensive. Will be used in final guide alongside (ii) options to build site resilience; and (iii) options that directly intervene at sites. |
| **2.** | **Methods for *appraising* options** | | | | | | | **Comments on rejected steps** |
| 2.1 | Conduct first-pass option screening. | | | | + | + | √ | |
| 2.2 | Choose formal or informal method. | + | | + | | + | ✗ | Formal method is unable to monetise site value. |
| 2.3 | If informal chosen, assess options against resources/constraints. | | + | + | | | ✗ | Use more than just these two criteria. |
| 2.4 | Select assessment criteria from a generic list. | + | | + | | | ✗ | No generic list available for cultural sites. |
| 2.5 | Identify a set of *assessment criteria*; devise a *scoring system*; and then rank options. | + | | + | | | √ | |

2.5.1 Identified *assessment criteria*

| Criteria | Source | Questions put to stakeholders |
|---|---|---|
| 1. Cost efficiency | 1. Chambwera et al. (2014) | 1. "Is the option affordable?" |
| 2. Goal orientation | 2. Noble et al. (2014) | 2. "Does the option meet our goals?" |
| 3. Practicality | 3. Klein et al. (2014) | 3. "Does option require available skills & capacities?" |
| 4. Cultural appropriateness | 4. Adger et al. (2014) | 4. "Is the option 'proper way'?" |
| 5. Co-benefit provision | 5. Huq and Reid (2004) | 5. "Will the option benefit the community in other ways?" |
| 6. Timeliness | 6. Stafford Smith et al. (2010) | 6. "Can we implement the option in a short time frame?" |
| 7. Robustness | 7. Lempert et al. (2013) | 7. "Will the option work if climate change is worse than expected?" |

2.5.2 *Scoring system* for responses to questions put to stakeholders: 'Yes' = 2 pts. 'Possibly' = 1 pt. 'No' = 0 pts.

**Table 2.** Results for option identification and option prioritisation. **Option identification:** a tick (√) indicates that an option was identified by rangers or archaeological risk literature. "Option 1. Do nothing" was added retrospectively, as from the project's inception, rangers indicated adaptation to be a priority need. "Option 20. Communicate sites' vulnerability" was added retrospectively, in light of Rockman et al.'s (2016; published after workshop) considering it an adaptation option. **Option prioritisation:** numerals indicate ranger rankings. A cross (✗) indicates an identified option was rejected during screening of a preliminary list of options. "NI" (not identified) indicates that an option identified during the brainstorming exercise by one ranger group was not identified by one or more of the other ranger groups. Option 20 was ranked '1' retrospectively due to instigation/completion of video prior to workshop.

| Option Identification | Djelk/Kakadu | Ashmore (2005) | Barclay et al. (1995) | Dawson (2015) | Rockman et al. (2016) | Rowland (1992) | Cassar et al. (2005) | Option Prioritisation | | |
| --- | --- | --- | --- | --- | --- | --- | --- | --- | --- | --- |
| | | | | | | | | Djelk Men | Djelk Women | KNP Men/Women |
| **Direct intervention options** | | | | | | | | | | |
| 1. Do nothing | √ | √ | √ | √ | √ | √ | √ | ✗ | ✗ | ✗ |
| 2. Defend coast—*comprehensively* | √ | √ | √ | √ | √ | | | ✗ | ✗ | ✗ |
| 3. Defend coast—*less than comprehensively* | | | √ | √ | | | | | | |
| 4. Excavate—*exhaustively* | | √ | √ | √ | √ | | | | | |
| 5. Excavate—*less than exhaustively* | | | √ | √ | | | | | | |
| 6. Surface-documentation generally | √ | | | √ | √ | | | 2 | 1 | 4 |
| 7. Surface-documentation—3D modelling and augmented reality | √ | | | | | | | 1 | 1 | 2 |
| 8. Relocate cultural site | √ | | | √ | √ | | √ | ✗ | ✗ | ✗ |
| 9. Replicate cultural site | | | | | | | √ | | | |
| **Options building cultural site resilience** | | | | | | | | | | |
| 10. Improve resilience generally | | √ | | | √ | | √ | | | |
| 11. Give sites protective legal designation | √ | | √ | | | | | 2 | 1 | 1 |
| 12. Modify site structure (no integrity loss) | | | | | √ | | | | | |
| 13. Eradicate feral animals entirely | √ | | | | | | | 3 | 3 | 6 |
| 14. Stabilise feral numbers via harvest | √ | | | | | | | ✗ | NI | NI |
| 15. Fence sites against feral animals | √ | | | | | | | 4 | 2 | NI |
| 16. Conduct fire management at sites | √ | | | | | | | 3 | 4 | 6 |
| 17. Erect gates on roads—to block vehicles | √ | | | | | | | NI | NI | 1 |
| **Options building stakeholders' adaptive capacity** | | | | | | | | | | |
| 18. Consult stakeholders generally | | √ | | √ | √ | √ | √ | | | |
| 19. Introduce risk assessment system | √ | √ | | √ | √ | √ | √ | 1 | 1 | 2 |
| 20. Communicate the vulnerability of sites | √ | | | | √ | | √ | 1 | NI | 1 |
| 21. Establish partnerships | √ | | | | √ | √ | √ | 1 | 1 | 3 |
| 22. Share knowledge | √ | | | | | | | NI | 1 | NI |
| 23. Integrate options with local planning | | √ | | | | | √ | | | |
| 24. Give training to local stakeholders | √ | | | | √ | | | 2 | 4 | 1 |
| 25. Address governance issues | √ | | | | | | | 1 | 4 | 5 |
| 26. Build adaptive capacity generally | | | | | √ | | | | | |
| **Total options identified** | 17 | 7 | 6 | 9 | 13 | 4 | 9 | **Total options prioritised** | | |
| | | | | | | | | 11 | 11 | 11 |

## 2.3. Option Appraisal

In line with the method commended by two sources [26,27], a first-pass screening of the preliminary list was conducted with the aim of removing options considered, in retrospect, unworthy of formal appraisal.

Contrary to the method commended by three sources [23,25,27], a decision by participants as to whether to employ a formal or informal option appraisal system was not included (see Table 1, Item 2.2). Formal methods, such as cost–benefit analysis or multicriterion analyses, focus mainly on financial implications. They are most often employed in a top-down planning context in which local values are rarely considered [36]. In a bottom-up, participatory planning process, however, cost should be one consideration among other planning implications. Analysis of measures to ameliorate the consequences of loss or damage to social values is better served by a deliberative, qualitative approach informally ranging across multiple nonmonetary criteria [37]. This is especially important in a cultural heritage context. Attempts have been made to monetise individual cultural site value by, for example,

using willingness-to-pay determinants [38]. However, cultural sites with a pure, market-based value are usually those generating tourism revenue [39]. This applies to a fraction of the total number of cultural sites. An informal, qualitative approach to decision-making, as opposed to a formal approach, can be exceptionally effective, and, when there is limited information, produce better results than formal methods [40].

The identification of a set of assessment criteria for the appraisal method (see Table 1, Point 2.5.1) by the lead author offered the opportunity to develop a comprehensive list of criteria appropriate to cultural site adaptation that capture concerns central to climate change adaptation literature. Preselecting assessment criteria, as opposed to requiring rangers to select their own criteria during workshop testing of the appraisal method, enabled comparison between the results of independent testing.

Seven criteria were chosen as a result of the literature review. The seven chosen, and their corresponding questions to put to rangers, were as follows:

1.  **Cost efficiency ("Is the option affordable?").** Some adaptation options will be technically possible, but must be dismissed because the cost of their implementation is beyond current financial resources. Cost should therefore be assessed, but in the context of nonmonetary values [37].
2.  **Goal-oriented ("Does the option meet our goals?").** Options should be sought and appraised against the overall goals of stakeholders established during the 'framing' or scoping step of the adaptation planning process [41]. As Hinkler [25] states, the question "What are we adapting for?" (the desired outcome) is as significant as, if not more so, as the question "What are we adapting to?". In this way, adaptation focuses on people's capacity and willingness to respond [42].
3.  **Practicality ("Does the option require skills and capacities available to us?").** Human resources are fundamental to option implementation [43]. These include skills, information, leadership and management capacity [44]. Considering human resources opens up options that might have been dismissed if finances were the only consideration [11].
4.  **Cultural appropriateness ("Is the option 'proper way'?").** Culture shapes the relationship of society to the environment and is an important determinant of responses to risks [45]. Options consistent with social norms will be more acceptable to local stakeholders [46–48]. In an Indigenous context, traditional protocols affect cultural site management and require oversight by traditional owners [49].
5.  **Co-benefit provision ("Will the option benefit the community in other ways?").** Options with co-benefits should be sought out [41,50]—they are more likely to be implemented than those with a single benefit [51]. In an Indigenous land management context, 'win-win' options will complement natural resource management [52].
6.  **Timeliness ("Can we implement the option in a short time frame?").** Options that can be implemented in the short- to mid-term have advantages over those with long lead times [43]. The latter face greater uncertainty [53] and the danger of immobilising decision-makers and exacerbating psychological, social or institutional barriers [54].
7.  **Robustness ("Will the option work if climate change is worse than expected?").** Robust or 'low regrets' options satisfy stakeholder goals under different future climate scenarios [27,55]. They also have advantages when downscaled climate projections are nonexistent or highly generalised, e.g., Reference [56], which is often the case for remote locations. For similar reasons, stakeholders should also favour flexible options—those that can be implemented in stages or dismantled easily [57,58].

In order to rank options using the seven criteria outlined above, a simple scoring system was devised. As per findings by Carmichael et al. [6], keeping the scoring system simple made for ease of use. A matrix was constructed with options on one axis and assessment criteria on the other. Each option was given scores for each of the seven criteria in the following way: if rangers answered

'yes' to the question corresponding to a criterion, the option earned two points; 'possibly' earned one point, and 'no' earned zero points. The seven scores for each option were then added up to produce a total score for each option.

*2.4. Options-Analysis Workshops*

The preliminary options list was reviewed, added to and then appraised by rangers at three workshops: one conducted with Kakadu male and female rangers, a second with male Djelk rangers and a third with female Djelk rangers (for cultural reasons, Djelk Rangers separate the work of male and female rangers).

Djelk female and male rangers took part in separate workshops, reflecting separation that exists in their current work practice. The KNP ranger workshop was conducted with five rangers (three male, two female). The male Djelk ranger workshop was conducted with 15 Indigenous rangers and one non-Indigenous ranger coordinator (female). The female Djelk ranger workshop was conducted with three Indigenous rangers, the total number of female rangers. The female Djelk rangers had not been involved in the scoping and risk analysis steps of the project, due to their being unavailable at the time research into those steps was conducted. The workshops lasted 61 min (KNP rangers), 58 min (male Djelk rangers) and 72 min (female Djelk rangers). Five one-on-one interviews were also conducted, lasting 30 to 45 min each. These were focused on complementing the numerical scoring of options with further Indigenous commentary on options.

Workshop participants and individual interviewees were either self-selected or invited to participate on the basis of their contribution during previous research phases. Participants' knowledge of English was generally good, so a translator was not necessary. Option workshops and interviews were transcribed and organised digitally according to activity and date, using the NVivo software package. Strategies to manage potential biases in data collection included reporting back to participants, and participants reviewing manuscripts and quotes within them presented as verbatim.

## 3. Results

The preliminary list of adaptation options for both climate change and nonclimate hazards (see Table 2) was as follows:

- defend the coast (i.e., comprehensively, with sea walls);
- surface documentation of high-risk sites generally for a local museum or database;
- relocate cultural sites;
- give sites protective legal designation;
- eradicate feral animals, in particular water buffalo (*Bubalus bubalis*);
- fence sites against feral animals;
- conduct fire management at sites;
- introduce a routine risk assessment and monitoring program by digitising the risk field survey (tested during previous steps in the research) and making it available on rangers' I-Tracker GPS data-collection tablets [59];
- establish partnerships with archaeologists and regional stakeholders;
- give training to local stakeholders;
- address governance issues.

During the brainstorming exercise, female Djelk rangers added "share knowledge" to the option list; male Djelk rangers flagged a recent proposal made by non-Indigenous enterprises to not eradicate buffalo but, instead, to "stabilise feral numbers via a harvest" for commercial gain; and KNP rangers added the option to erect more "gates on roads—to block vehicles" driven by tourists.

At all three workshops, an additional option was proposed by the lead author: developing an augmented reality application (as distinct from a virtual reality application) to allow 3D models of

sites, particularly rock art, to be re-experienced once lost via an augmented reality ocular headset on their original, non-virtual rock face (where access to the site remains possible).

A Djelk ranger summarised the dilemma in the following way:

*The Djomi museum* [local museum in the township of Maningrida] *is really good, taking photos and getting information, but in my way, I want to see it 'live'; paintings, right there!*

No such application currently exists, so the application of the augmented reality concept in this regard is hypothetical. Open to the idea, workshop participants watched a promotional video, viewable at link: https://www.youtube.com/watch?v=xXy7lbs-D48&sns=em [60], for the then newly released Microsoft Hololens® augmented reality ocular headset. The initial first-pass screening process saw options such as "defend coast" and "relocate cultural site" dismissed by all groups as impractical, too costly and culturally inappropriate. As a Djelk Ranger declared:

*Sea walls? Nah! The sea is a really big thing, you can't do anything like that. The sea level is coming up and* the *floodplain will be filled up, you can't do anything about this.*

Buffalo harvesting was also rejected, due to the impacts a maintained herd and capture vehicles would have on natural values and cultural sites.

Of those options identified by all ranger groups, appraisal via the assessment criteria resulted in six being ranked by one or more group as a primary priority (i.e., priority '1', see Table 2). These were cultural site documentation via 3D photogrammetry for an augmented reality app, introduction of a risk management system, establishing partnerships, providing training, giving sites protective legal designation and addressing governance issues.

A seventh option, "communicate the vulnerability of sites" was not considered during the construction of the preliminary options list. Rockman et al.'s work [32], published immediately after the options analysis workshops were conducted, considers communication of threats to cultural sites to be an adaptation planning option in its own right. In light of Rockman et al.'s strategic recommendation, and a similar position taken by Cassar and Pender [61] under the rubric of "education", we acknowledge that the early stipulation by rangers (when initially agreeing to take part in this research project) that a documentary video be an additional output of research (in order to raise popular awareness of the vulnerability of cultural sites to climate change and elicit support) constituted identification of this option. Given that ranger identification of the option also stipulated its immediate implementation, and that the video was largely complete prior to the option workshops, we considered it to have been effectively prioritised at the highest level by rangers, and therefore retrospectively assigned it to priority level '1' (see Table 2, above).

The following sections present rangers' and community members comments on high-scoring options.

### 3.1. Cultural Site Documentation via 3D Photogrammetry for Augmented Reality

While coastal protection and relocation were dismissed, new technological approaches to surface documentation of sites were enthusiastically endorsed. Rangers considered making 3D models of the most vulnerable riverine rock art sites and viewing these, once the original was lost, at their original locations (where practical) via an augmented reality ocular headset. A KNP ranger noted that:

*There's no problem with that so long as we have a little bit of help. It could bring everything back to life, we can make a record that will be there forever – that new technology could help.*

For both ranger groups, however, cultural protocols, consultation with traditional owners and control over imagery would have to be very strictly maintained.

### 3.2. Introduction of a Risk Assessment System

Rangers conceived numerous benefits of digitising the risk field survey for their GPS-based field monitoring devices. In addition to mainstreaming the monitoring and prioritisation of cultural sites, digitisation would produce data allowing financially deficient rangers to canvass for more funding and increased managerial support for extended site maintenance. As a KNP ranger pointed out:

> *Rangers have to adapt it* [the Risk Field Survey] *into the routine conservation checks. It's important to know what you're dealing with and what is important before you go and push other people to help you.*

### 3.3. Communicate the Vulnerability of Cultural Sites

From the project's outset, formally communicating the problem of cultural site vulnerability to climate change was flagged as urgent. Rangers felt that non-Indigenous Australians were not listening to their repeated warnings about the threat of climate change. Work on the documentary, *Places in Peril: Archaeology in the Anthropocene* (viewable at link https://vimeo.com/203773921) therefore began immediately [8]. As one KNP ranger explained during initiation of the documentary project:

> *Climate change is going to be a big thing throughout Australia. A video is definitely the way to go . . . It will help people better understand climate change* [impacts on cultural sites] *as well; a lot more other groups will want to start getting involved.*

### 3.4. Partnerships

All groups stressed the value of partnerships, including those with archaeologists, cultural heritage managers and Indigenous consultative agencies such as the Northern Land Council (NLC). Ultimately, however, partnerships were downgraded by KNP rangers because they were perceived as all too often lacking financial backing. As one explained:

> *Few partners come up with the money for all the stuff to do with sacred site maintenance. You need money. Anyone could become a partner, but once you start mentioning funding, no one wants to put their hand up.*

### 3.5. Training

Training was considered vital for using the risk field survey, general rock art recording and maintenance, and in following cultural protocols. A KNP ranger noted that:

> *Training meets our goals, and it's in the* [Kakadu National Park] *Plan of Management, which says that we are supposed to protect rock art. Training needs to include cultural protocols. You can't have people looking at* [i.e., working at] *the sites that don't know what they're doing.*

### 3.6. Giving Sites Protective Legal Designation

Barclay et al. [29] listed protection through legal designation as an important coastal protection measure. Rangers independently concurred; listing more sites with the Aboriginal Areas Protection Authority (AAPA), an independent statutory body charged with overseeing the protection of Aboriginal sacred sites on land and sea in the Northern Territory [62], would have additional benefits. As a Djelk ranger pointed out:

> *AAPA needs to come and work with Traditional Owners. It would be really good, I think, if AAPA registered all the sacred sites; that would give us more power to stop mining and stop people coming in looking for oil, gas and the like.*

*3.7. Governance*

Djelk and KNP rangers had different views on governance. For KNP rangers, there was an issue with a lack of consultation around resource provision for maintenance of cultural sites other than those open to tourists. They ranked this issue down, however, because they did not feel confident of achieving results. Ultimately, control was seen to rest with the Australian government. As a KNP ranger said:

> *How are you going to change the policies? In the 1980s, Kakadu was the place to visit. So we had a lot of money, and a lot of staff to look after a lot of different areas. But today the Park is getting no revenue. At the end of the day, it depends who plays politics best and gets in* [i.e., who wins a federal election].

Djelk rangers felt confident about resisting interventions by non-Indigenous administrative staff working for their parent Aboriginal corporation. Raised as a live issue during the scoping step, by the time of the options analysis workshop it had been resolved by way of non-Indigenous administrative staff changes. As noted by a Djelk ranger:

> *We are working together now, we have solved that problem. Office mob* [non-Indigenous administrative staff] *have now put it* [intervening in Djelk natural resource management planning] *on the side. They are focusing more outside of ranger stuff now.*

## 4. Discussion

Rangers initiated and considered more options than any one of the other studies considering adaptation of cultural sites cited in Table 2 [7,28–32]. Our study underscored, therefore, the value of locally controlled planning—direct experience of risks proved highly fertile ground for appraising practical measures. It also underscored the value of Indigenous traditional knowledge and experience [63,64] to the adaptation of cultural sites, and the extent to which this experience can complement professional cultural site adaptation. Local control might therefore be considered an adaptation option in its own right, one made demonstrably feasible by this study. Our study also underlined the value of collecting options identified informally throughout preceding phases of the adaptation pathway [23,26,27], or in other words, not siloing the options-analysis process.

We distinguished three classes of option among the list identified by rangers: (1) direct intervention options, such as defending the coast or surface documentation of sites in the face of inevitable loss; (2) cultural site resilience-building options, such as giving sites legal protection; and (3) adaptive-capacity building options, such as digitising the risk field survey. The following sections discuss these classes.

*4.1. Direct Intervention*

There are often clear limits to climate change adaptation [65], and studies of Indigenous stakeholders planning whole-of-community climate change adaptation within the context of sea level rise have reported both important limits as well as cultural barriers [66]. Our study, however, documents Indigenous rangers confronting inevitable loss or damage to cultural values with a high degree of pragmatism and few cultural constraints. Rangers were prepared to bear losses and ameliorate the consequences of loss with surface documentation, while at the same time exploring new technological aids to this end. Climate change adaptation options that use new technology can potentially benefit disadvantaged populations [41,67].

Reilly [68], an early exponent of virtual reality as an archaeological research tool, referred to 'virtual archaeology' as the modelling of landscapes, excavations, buildings and artefacts with computer applications in order to test scientific questions, but also to communicate the past to nonspecialists. Digital 3D imagery and visualisation has since become available for many iconic archaeological sites globally [69], for Australian Indigenous cultural sites, e.g., References [70,71], and even for lost

sites—for example, the Temple of Bel in Palmyra, Syria [72]. While virtual reality provides an immersive experience of a cultural site at an alternative location, emergent augmented reality hardware that supplements reality with 3D imagery might allow Indigenous traditional owners to secure threatened traditional cultural knowledge but also location-dependent, traditional cultural practice. Experiencing a lost cultural site in its original location allows users to maintain their own, and a site's, "connection to Country". Connection to Country is essential to Indigenous cultures—land, language and place are embedded in kinship relations, identity, belief systems, justice codes, spirituality and Indigenous sovereignty, as well as physical, social and emotional wellbeing [73].

While broadly supportive of an augmented reality option, rangers expressed some concerns about the implications of producing, regulating access to, and storing a proxy for a lost site. Aside from their identified need for community consultation prior to recording, other issues arise: ensuring that culturally sensitive imagery does not find its way, via the internet, into the public domain, where it can be appropriated and altered, and ensuring that GPS-connected cameras do not reveal site locations [74]; ensuring that the chosen repository allows culturally appropriate equity and ease of access to content [75]; ensuring that a repository does not enter into sharing agreements with other less secure organisations, or that the use of imagery by financially disadvantaged communities for tourism or other income-generating opportunities does not have unintended outcomes [76].

### 4.2. Building Cultural Site Resilience

Building cultural site resilience to climate change is an option not well represented in cultural site climate change adaptation studies. Rangers' holistic understanding of landscape-scale processes conceived of measures such as culling buffalos as a climate change adaptation option. Their insights are consistent with IPCC [63] findings recognising that invasive species can benefit from climate change, due to a decline in competition from less resilient native species, and increase landscape-scale vulnerability to climate change because of the environmental degradation they cause [63,77].

Climate change impacts can exacerbate existing vulnerabilities. Cultural sites that are well maintained and resilient to impacts not related to climate change will therefore be less vulnerable to climate change impacts. Adaptation planning that considers both climate and nonclimate impacts, such as legal protection or gates to restrict tourist access, represents the 'mainstreaming' of climate adaptation into broader risk management, which increases the likelihood of actual implementation of an adaptation plan [50,78]. A program documenting vulnerable cultural sites is essential in this instance, as Northern Territory legislation (Northern Territory Aboriginal Sacred Sites Act and Northern Territory Heritage Act) already provides legal protection to known sites, whereas unrecorded sites remain at risk.

### 4.3. Building Local Adaptive Capacity

Our results highlight the importance of local adaptive capacity. Communicating the vulnerability of cultural sites to climate change was cited as an option by only two studies [31,32], but seen by rangers as one of the most critical. Indeed, distribution of the resulting documentary film elicited contact from science journalists and entrepreneurs inspired to cover 'the story' or support implementation of options such as 3D modelling of sites and augmented reality application development.

Unlike any other cultural site adaptation study, we considered governance issues. Rangers working directly for the government (KNP Rangers) had low confidence in their ability to address governance barriers. Wider research stresses that low Indigenous involvement in formal, government decision-making processes regarding resource allocation decreases resilience [79]. Transformational change, or adaptation involving a new governance model or a fundamental shift in power [65,80] might ultimately benefit the KNP Rangers. In contrast, the Djelk Rangers, operating under the auspices of an Aboriginal corporation, were able to successfully address issues around planning autonomy.

Equipped with the risk field survey, rangers might offer heritage management training and consultation services on a fee-for-service basis, regionally, nationally and even internationally [4].

Providing market services in relation to customary values fits with Altman's [81] hybrid economy model for Indigenous development. Here, Indigenous landowners derive economic benefit from three sources: market, state and customary economies.

Ranger prioritisation of partnerships with archaeologists is particularly significant. The risk field survey assesses risk to sites partly via a ranking of cultural value. However, rangers recognise that archaeologists might potentially augment the power of their assessment results via a complementary scientific assessment of significance. Indeed, archaeologists are increasingly recognising the value of significance assessment derived from cultural value [82,83].

### 4.4. Prompts and Generic Lists

In light of our identification of three option types, we envisage adding a prompt to the process of option identification. Some generic adaptation option guides [24–27] commend specific prompts. We dismissed these prompts as insufficient or incomplete. Our proposed prompts, however, ask: "What can we do for sites that are in danger of being lost to climate change?"; "What can we do to keep sites strong and healthy?"; and "What can we do to make ourselves more able to make sites strong against climate change?".

The options identification phase of the process employed here did not present rangers with a generic list of options from which to choose. Future options analysis, however, can now do so. This study can now be used to generate a generic list comprising options identified by rangers as well as those gleaned from academic and managerial studies. When data have accumulated from routine risk field survey application, options rejected during this study by Djelk and KNP rangers might conceivably become tenable to them. For example, some form of coastal protection or relocation—options which have been undertaken by citizen archaeologists in the UK and US [84,85]—may become favoured options for as-yet-unconsidered sites.

## 5. Conclusions

This study tested an options-analysis method for Indigenous cultural site adaptation. It did so by facilitating its use with two Indigenous ranger groups to determine whether it could elicit meaningful and comprehensive responses, and whether Indigenous rangers had the organisational capacity to fulfil the requirements of the process. It found that the rangers were highly engaged by the approach and were able to supply detailed and considered responses born of direct observation and insightful appreciation of the climate challenges confronting their cultural sites. In testing a methodology for participatory cultural-site options analysis, our study identified an unprecedented range of adaptation options.

The multiple insights and outcomes from our study include a repeatable and transferable process with applicability elsewhere in Australia, and with the potential to provide systematic guidance for cultural site climate change responses internationally. Although the participants in the study were Indigenous rangers, no element of the method excludes its use by non-Indigenous stakeholders. The approach facilitates reflexive learning and continued iteration, with the implication that new, locally specific adaptation options will continue to be identified. International application of the method by local and Indigenous users might see these new additions shared among a global community of cultural site adaptation practitioners.

The development of global strategies to combat climate impacts on cultural heritage sites has stalled since the first steps were taken at the beginning of the current millennium [86]. While renewed efforts need to facilitate the adoption of risk analysis across global, regional and local scales, the subsequent risk management step of options analysis should also be pursued. The fostering of links needs to be developed between those working independently at a local scale in order to share knowledge born of empirical experience.

**Author Contributions:** B.C. was responsible for conceptualization, formal analysis, investigation and final write up; G.W., I.N., S.N. and J.C. were responsible for investigation; D.B. and C.D. were responsible for review and

editing; B.W. and S.B. were responsible for supervision. All authors have read and agreed to the published version of the manuscript.

**Funding:** Fieldwork was supported by the Australian Research Council (Linkage Project LP110201128 and Discovery Project DP120100512), the Australian National University and Charles Darwin University.

**Acknowledgments:** This research was conducted in Kakadu National Park under Permit No. RK854 and in the Djelk IPA under NLC Permit No. 57159. The authors wish to thank the following staff: at Parks Australia, Simon Dempsey, Natasha Nadji, Jeffrey Lee, Bobby Maranlgurra, Kadeem May, Gabrielle O'Loughlin; and at the Djelk IPA, Darryl Redford, Obed Namirrik, Alfie Galaminda, Bobbie-Sheena Wilson, Felina Campion, Dominic Nicholls, Alys Stevens, Anthony Staniland and Ricky Archer. Critical feedback was received from Apolline Kohen and Rolf Gerritsen. Fieldwork was supported by the Australian Research Council (Linkage Project LP110201128 and Discovery Project DP120100512), the Australian National University and Charles Darwin University. The findings and views expressed are those of the authors and do not necessarily represent the views of Parks Australia, the Director of National Parks, the Australian Government, Djelk Rangers or Bawinanga Aboriginal Corporation.

**Conflicts of Interest:** The authors declare no conflict of interest.

**Compliance with Ethical Standards:** The study followed standard ethical norms, including obtaining university ethics approval (Australian National University No. 2014-342, Charles Darwin University No. H14022), eliciting informed consent from all study participants, reviewing results with and presenting results back to communities prior to publication and not divulging the locations of 'sacred' sites.

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
