# Peer review of "A Methodology for the Assessment of Climate Change Adaptation Options for Cultural Heritage Sites"

_climate, doi:10.3390/cli8080088_

Round 1

Reviewer 1 Report

The article novelty lies  in, testing and use of novel approach in adaptation of cultural heritage sits to climate change . However, structural weakness in the organisation interfere with the flow. The discussion and conclusion need to be improved. In particular the segmentation of the methodology section need to be addressed. 

Section

Observation

Abstract/ title and references

Abstract need to be redone

Check the incomplete references

Reorganize  the title

See highlighted text

Introduction

Mixed up with methods. Need to be improved further

See highlighted text

Methods

Though adequately explained, there is mix up in several sections hence repetitions,. Reorganize he write up

See highlighted text

Results

Need to synthesize the findings

See highlighted text

Discussion and Conclusion

Limitations not well expounded

Line 424 need to revised as it is not possible for the organisational and planning ability of the respondent to have been established during the research activity

The conclusion statement( Line 538 is abrupt. Hence the anchor statement  need to capture the study objective first

See highlighted text

Author Response

Thank you for your review.

  • The abstract has been now been amended
  • Incomplete references have been amended
  • The title has been recast
  • The Introduction has been reorganised
  • The Methods section has been reorganised
  • The Results section has been reorganised
  • The conclusion has been reorganised

Again, thank you.

Reviewer 2 Report

Lines 70 to 81 need to be better illustrated the analysis methodology and after the case studies in a more exhaustive way

Lines 538 to 551 the conclusions are not clearly written and should be improved by explaining how the methodology can be exported to other contexts

The bibliography should be updated with the most updated texts after 2017

Author Response

Thank you for your review.

  • Lines 70 to 81 have been amended.
  • Lines 538 to 551 and the conclusion have been amended
  • The bibliography has been updated with texts published after 2017.

Again thank you.

Author Response

Thank you for your review.

A methodology for the assessment of options for the adaptation of cultural heritage to climate change

>> play a significant role in community identity, cohesion and sense of place, and this is particularly the case for Indigenous peoples (Keen, 2004).

It should be noted that these places often provide a connection and/or a foundational block for maintenance of cultural practices and continuity of cultural behaviors.

AUTHOR: THANKS – point now made in text

>> have perceptions of sea level rise and more extreme storm surge increasing erosion of coastal middens and floodplain-fringing middens and rock art; more intense cyclones are perceived to be impacting coastal middens; and more extreme and frequent precipitation events are perceived to be eroding inland riverine rock art and contributing to the erosion of floodplain-fringing middens and rock art (Carmichael, 2015).

“Perceived” twice, suggest using an alternative for one of these

AUTHOR: THANKS – change made

This would be a nice section to add what their “perception” is, is the utilization of Traditional Ecological Knowledge (TEK) that is honed

AUTHOR: THANKS – the text lists their perceptions.

>>Indigenous rangers undertake natural and cultural resource management, including fire, feral animal and weed management, cultural site maintenance and tourist liaison.

It would be nice to read how they do this – do they employ Traditional Knowledge information form the local Indigenous? Are they relying on their own set of skills? Is there a TEK management system in place with help of Indigenous?

AUTHOR: THANKS – the text now states: Their management practice draws extensively on traditional environmental knowledge, with a large proportion of Indigenous rangers being Traditional Owners of domains within the Park.

>> Djelk is directed by an Indigenous executive committee.

This is important because.... ?  (they help guide the policies that manage, or they assist with cultural decision making or etc – help the readers understand why having Indigenous-oversight is vital to maintenance of cultural sites)

AUTHOR: THANKS – the text now states: As a subsidiary of Bawinanga Aboriginal Corporation, Djelk is directed by an Indigenous executive committee, which ensures all decisions respect Indigenous cultural protocols.

>> but is difficult to perceive by those managing natural resources over the short to medium term

Excellent explanation to someone who wouldn’t know, though contrasting that with information about how TEK is a long-term practice via Indigenous groups

AUTHOR: THANKS for the complement.

>> They are most often employed in a top-down planning context in which local values are rarely considered

Excellent to point out – with this approach, often are Indigenous voices who are not included as well. 

AUTHOR: THANKS for the complement.

>> Analysis of measures to ameliorate the consequences of loss or damage to social values is better served by a deliberative, qualitative approach informally ranging across multiple non-monetary criteria (Chambwera et al., 2014). This is especially important in a cultural heritage context.

YES.

AUTHOR: THANKS for the complement.

>> one conducted with Kakadu male and female rangers, a second with male Djelk Rangers, and a third with female Djelk Rangers

Was there cultural reasoning for the separation of rangers, or was it coincidental? Issues in Indigenous culture where decisions are made often play a part in issues, and/or communication – this would help readers better understand why the divisions occurred, whether intentional or not.

AUTHOR: THANKS – the text now states: Djelk female and male rangers took part in separate workshops reflecting separation that exists in their current work practice.

>> For both ranger groups, however, cultural protocols, consultation with Traditional Owners and control over imagery would have to be very strictly maintained

Excellent to see Indigenous data sovereignty addressed

AUTHOR: THANKS for the complement.

>> Ultimately, control was seen to rest with the Australian Government

>> felt confident about resisting interventions by non-Indigenous administrative

 staff working for their parent Aboriginal Corporation

Good to see this issue (control) addressed directly/overtly and illustrating the juxtaposition in this study so perhaps more attention and power can be brought to support Indigenous groups and their knowledge

AUTHOR: THANKS for the complement.

>> might allow Indigenous Traditional Owners to secure threatened traditional cultural knowledge but also location-dependent, traditional cultural practice. Experiencing a lost cultural site in its original location allows users to maintain their, and a site’s, connection to Country. Connection to Country is essential to Indigenous cultures – land, language and place are embedded in kinship relations, identity, belief systems, justice codes, spirituality and  Indigenous sovereignty, as well as physical, social and emotional wellbeing (Ganesharajah, 2009).

All of this relates to TEK, I’m surprised none of it is related or cited

AUTHOR: THANKS – text contains a citation.